# Apolipoprotein A-IV: A Multifunctional Protein Involved in Protection against Atherosclerosis and Diabetes

**DOI:** 10.3390/cells8040319

**Published:** 2019-04-05

**Authors:** Jie Qu, Chih-Wei Ko, Patrick Tso, Aditi Bhargava

**Affiliations:** 1Department of Pathology and Laboratory Medicine, Metabolic Diseases Institute, University of Cincinnati, 2180 E Galbraith Road, Cincinnati, OH 45237-0507, USA; quje@ucmail.uc.edu (J.Q.); koce@ucmail.uc.edu (C.-W.K.); tsopp@ucmail.uc.edu (P.T.); 2Department of Obstetrics, Gynecology & Reproductive Sciences, University of California, 513 Parnassus Avenue, San Francisco, CA 94143-0556, USA

**Keywords:** apolipoprotein A-IV, apoA-IV-interacting proteins, lipid metabolism, atherosclerosis, platelet aggregation and thrombosis, glucose hemostasis, food intake

## Abstract

Apolipoprotein A-IV (apoA-IV) is a lipid-binding protein, which is primarily synthesized in the small intestine, packaged into chylomicrons, and secreted into intestinal lymph during fat absorption. In the circulation, apoA-IV is present on chylomicron remnants, high-density lipoproteins, and also in lipid-free form. ApoA-IV is involved in a myriad of physiological processes such as lipid absorption and metabolism, anti-atherosclerosis, platelet aggregation and thrombosis, glucose homeostasis, and food intake. ApoA-IV deficiency is associated with atherosclerosis and diabetes, which renders it as a potential therapeutic target for treatment of these diseases. While much has been learned about the physiological functions of apoA-IV using rodent models, the action of apoA-IV at the cellular and molecular levels is less understood, let alone apoA-IV-interacting partners. In this review, we will summarize the findings on the molecular function of apoA-IV and apoA-IV-interacting proteins. The information will shed light on the discovery of apoA-IV receptors and the understanding of the molecular mechanism underlying its mode of action.

## 1. Introduction

Lipids, such as cholesterol, phospholipids, and triglycerides (TG), are insoluble in water. Therefore, their transport in the blood, lymph, and extracellular fluid requires association with proteins that have the capacity to interact with both lipids and water. Lipoproteins are complex particles comprised of a central core of cholesteryl esters and TG surrounded by an outer layer of free cholesterol, phospholipids, and apolipoproteins [1]. Apolipoprotein A-IV (apoA-IV) was originally identified as a major protein component of chylomicrons in postprandial lymph and in the plasma of both human and rats [2,3,4]. Since its discovery 40 years ago [4], numerous research using cell cultures and animal studies has been carried out to determine the physiological roles of apoA-IV. To date, apoA-IV is known to participate in a broad spectrum of biological processes, including lipid metabolism [5,6,7], reverse cholesterol transport [8,9,10], protection against atherosclerosis [11,12,13], platelet aggregation and thrombosis [14], glucose hemostasis [15,16,17,18], as well as food intake [19,20]. In human clinical studies, low plasma apoA-IV concentration is associated with coronary artery disease in men [21]. High apoA-IV pre-surgical levels are related to improved insulin sensitivity after Roux-en-Y bypass surgery, which is an effective treatment for obesity and obesity-related co-morbidities, such as diabetes [22]. High plasma apoA-IV levels are also correlated with mild to moderate kidney failure, thereby acting as a predictor for kidney disease progression [23,24]. Furthermore, apoA-IV can serve as early diagnostic biomarkers for prediabetes, liver fibrosis, and ovarian cancer [25,26,27]. Given the protective role against atherosclerosis and diabetes as well as a yet unidentified role in kidney disease, apoA-IV may become a new and effective therapeutic target for the treatment of these diseases. To achieve this goal, it is of paramount importance to understand the molecular mechanisms by which apoA-IV exerts its regulatory function. Notably, except platelet integrin αIIbβ3, the receptor for apoA-IV in other cell types remains elusive [14]. In this review, we will first summarize our current understanding of *APOA4* gene, apoA-IV protein structure, apoA-IV glycation, and apoA-IV isoforms. Second, we will discuss the cellular processes in which apoA-IV is likely involved and the signaling event it may trigger by focusing on lipoprotein metabolism, reverse cholesterol transport, anti-atherosclerosis, platelet aggregation and thrombosis, glucose homeostasis, and food intake. Finally, we will conclude with a model describing pathways to which apoA-IV may contribute based on our current knowledge and provide some perspectives on future directions.

## 2. *APOA4* Gene and ApoA-IV Proteins

The *APOA4* gene is located on chromosome 11 in human and chromosome 9 in mouse. In humans, it is part of the gene cluster containing *APOA1*, *APOC3*, and *APOA5* [28]. From 5′ to 3′ the gene order is *APOA1*, *APOC3*, *APOA4*, and *APO5*, with the *APOC3* gene transcribed from the opposite DNA strand [28]. *APOA4* is located 12 kb downstream of *APOA1* and 28 kb upstream of *APOA5* [28,29]. It consists of two introns and three exons spreading over 2.6 kb. The first intron separates most of the apoA-IV signal peptide from the N-terminal mature apoA-IV protein, and the second intron divides a highly conserved, variant amphipathic peptide repeat from the rest of the mature protein [30]. Genome-wide association (GWAS) meta-analysis from three independent cohorts (combined *n* ≥ 5000) identified three independent single nucleotide polymorphisms (SNPs) from two genomic regions that were significantly associated with apoA-IV concentrations; rs1729407 near APOA4, rs5104 in APOA4, and rs4241819 in Kallikrein B (KLKB1) [31]. The lead SNP rs1729407 was located between *APOA5* and *APOA4* and one missense variant (rs5104). The missense variant rs5104 results from a serine to asparagine substitution (Ser147Asn). Interestingly, the lead SNP rs1729407 showed an association with HDL-cholesterol, whereas the missense SNP, rs5104 has been shown to associate with dyslipidemia in Han Chinese [32], with postprandial apoA-I plasma concentration in healthy young men [33], and with triglyceride in response to lipid- lowering treatment drug, fenofibrate [34]. No apoA-IV SNP showed interaction with sex [31].

The *APOA4* gene encodes a 46 kilodalton precursor of 396 amino acids, which undergoes post-translational cleavage to yield a mature protein of 376 amino acids. It is primarily expressed in mammalian intestine and a small portion in rodent liver [35,36]. The majority of apoA-IV production occurs in jejunum, and it is also expressed in duodenum and ileum [37]. Additionally, it is detected in hypothalamus and the solitary tract of the brainstem as well as dendritic cells of the immune system [38,39,40]. In small intestine, *APOA4* is induced by active lipid absorption. ApoA-IV protein is synthesized in enterocytes and incorporated onto nascent TG-rich chylomicrons that are secreted into intestinal lymph and finally drained into circulation through thoracic duct. During subsequent hydrolysis of TG by lipoprotein lipase, most of apoA-IV dissociates from chylomicron particles. This dissociation may be attributed to: (1) The shrinking surface area; (2) the changes in the surrounding environment of the particles; (3) the competition for lipid binding by other exchangeable apolipoproteins E and C (apoE and apoC). Approximately 25% of apoA-IV transfers to high-density lipoproteins (HDL) and the rest exists in free form in the plasma [41,42].

### 2.1. ApoA-IV Protein Structure and Post-Translational Modifications

Human apoA-IV contains highly conserved, 22-amino-acid peptide repeats that can form amphipathic α helices [43]. In the absence of lipids, apoA-IV self-associates to form homodimers. Each monomeric apoA-IV protein is composed of a well-structured central core and less organized N- and C-termini [44]. The central helices (amino acids 64–335) are grouped into 4-helix bundles, helix-A (aa 74–94), helix-B (aa 96–204), helix-C (aa 206–255), and helix-D (aa 265–312) [43]. Helix-B is the longest helix and configures a rod-like structure. The N-terminal helix-A is perpendicular to helix-B and caps one end of the rod. Following a 180° turn to B, helix-C and D fold into an anti-parallel helical arm spanning approximately half the length of helix-B and are located at the other end of the rod. The central region of the two monomers are aligned in an anti-parallel fashion with the loops connecting C and D in each monomer opposing each other at the center of the rod. This interlocking orientation allows the burial of hydrophobic surface on one monomer by the other at the dimer interface. It is proposed that lipid is initially inserted into the central hydrophobic pocket of the dimer, and as more lipids are incorporated, the helical arms consisting of helix-C and D relax away from helix-B backbone, resulting in the formation of a circular ring and ultimately a disc-like lipoprotein particle [43]. The N-termini and C-termini from the opposing monomer confer intermolecular interaction at one end of the rod. They play critical roles in manipulating lipid-binding capacity of apoA-IV [44,45,46,47].

In the circulation, human apoA-IV was detected as a glycoprotein, which contains 1.8% mannose, 1.55% galactose, 1.55% N-acetyl glycosamine, and 1.1% sialic acid by weight [48]. The degree of non-enzymatic glycation of circulating apoA-IV correlated with the severity of coronary artery disease in diabetes patients [49]. Recombinant glycated apoA-IV (g-apoA-IV) prepared from glyoxal culture was shown to stimulate inflammatory reactions in vitro and in vivo. Additionally, g-apoA-IV enhanced the development of atherosclerotic lesion in apoE deficient mice that are prone to atherosclerosis [49]. Mutation in major glycation sites dampened the atherogenic effect of g-apoA-IV. Together, it suggests glycation of apoA-IV adversely influences its anti-atherosclerosis function.

### 2.2. Genetic Variants of ApoA-IV in Human Population

In humans, seven genetic variants have been identified, including apoA-IV-1, apoA-IV-1A, apoA-IV-2, apoA-IV-2A, apoA-IV-3, apoA-IV-0, and apoA-IV-5 [50]. ApoA-IV-1 is the most common isoform. ApoA-IV-1A encodes a T347S substitution near the C-terminus and has the same electrophoretic mobility as the parent isoform apoA-IV-1. It may facilitate the intravascular clearance of TG-rich lipoproteins at postprandial state, likely due to its reduced lipid-binding affinity based on structure prediction [51]. Several population studies reported lower plasma total cholesterol, LDL, and lower lipoprotein levels in carriers of the T347S allele than those homozygous in 347T [50]. ApoA-IV-1A was also associated with lower plasma apoA-IV concentration and increased risk of coronary heart disease [52]. ApoA-IV-2 is derived from a Q360H substitution at the C-terminus, introducing one more basic charge unit to apoA-IV-1. ApoA-IV-2 protein displays higher lipid affinity than apoA-IV-1. In contrast to apoA-IV-1A, it delayed the postprandial clearance of TG-rich lipoproteins [53]. Many population studies showed that people with the Q360H allele have increased HDL cholesterol, reduced LDL cholesterol, lower fasting plasma TG, lower apoA-I level, and higher fasting glucose and insulin levels [50]. Conversely, some other studies did not observe any effect on plasma lipids and lipoproteins. Although apoA-IV-2 cannot act as an independent risk factor for cardiovascular diseases in patients with normal lipid levels [54], it could promote atherosclerosis in patients with hyperlipidemia resulting from metabolic disorders, such as obesity, or familial combined hyperlipidemia [55,56]. ApoA-IV-1, apoA-IV-1A, and apoA-IV-2 are the top three frequent isoforms in human populations worldwide. A highly conserved domain containing a repeated series of EQ(Q/A/V)Q-type tetrapeptides is present at the C-terminus (aa 340-370) of mammalian apoA-IV proteins [57]. ApoA-IV-2A has a deletion at the second EQQQ motif at position 361, whereas apoA-IV-0 and apoA-IV-5 both encode an insertion of EQQQ motif at position 361 with apoA-IV-5 possesses additional G to T substitution at codon 317. ApoA-IV-3 is a two-charge shifted isoform. Polymorphisms that code for E24K or E165K or E230K substitution have been reported as apoA-IV-3 [51,58,59].

## 3. ApoA-IV Function in Cells

### 3.1. ApoA-IV and Lipoprotein Metabolism

ApoA-IV plays a role in modulating TG-rich lipoprotein assembly and metabolism. Of the major apolipoproteins produced by the gastrointestinal tract (apoA-I, apoA-IV, and apoB48), apoA-IV is the most responsive to lipid ingestion [60,61]. The induction of apoA-IV synthesis is mediated through absorption of long-chain fatty acids, but not short-chain fatty acids, which are transported via portal blood and do not elicit chylomicron production [62]. Upon lipid feeding, apoA-IV is assembled into nascent chylomicrons in the intestinal enterocytes [63]. To understand apoA-IV’s role in lipid absorption and packaging, the Black group transiently expressed apoA-IV in newborn swine intestinal epithelial cells IPEC-1 [6]. They found that apoA-IV overexpression caused a great increase in the secretion of TG, cholesteryl ester, and phospholipids in the chylomicron particles. They further expressed apoA-IV under an inducible promoter in IPEC-1 and showed that apoA-IV enhanced TG secretion in a dose-dependent manner, leading to the formation of larger TG-rich chylomicrons [64]. In the same study, they demonstrated that a region from residues 344 to 354 in human A-IV was essential for its ability to promote TG secretion while the C-terminal EQQQ-rich region played an inhibitory role. Apolipoprotein B is another key component of chylomicron particles and is critical for chylomicron assembly and secretion. Prechylomicron formation involves lipidation of apoB by microsomal triglyceride transfer protein (MTP) in the endoplasmic reticulum (ER) of enterocytes [65]. These prechylomicron particles, which also contain ApoA-IV on their surface, are later transported to Golgi apparatus, where they receive large amount of additional TG and expand their size before exocytosis [63,65]. When apoA-IV was restricted in the ER of McA-RH7777 rat hepatoma cells, apoB trafficking from ER to Golgi was dramatically reduced, resulting in decreased apoB secretion as well as TG secretion [7,66]. In contrast, over-expression of rat apoA-IV increased apoB secretion, enhanced apoB-containing particle expansion, and TG secretion [7,66]. This indicated that apoA-IV interacted with apoB in the secretory pathway to regulate TG-rich lipoprotein secretion. Interestingly, in vivo studies using apoA-IV knockout or transgenic mice revealed that alteration of apoA-IV levels did not change dietary lipid absorption [67,68]. However, loss of function of apoA-IV affected chylomicron assembly and metabolism as apoA-IV^-/-^ mice secreted larger chylomicrons and chylomicrons from apoA-IV^-/-^ mice were cleared more slowly from the plasma than WT chylomicrons [69].

ApoA-IV promotes lipoprotein lipase activity in the presence of apoC-II-containing lipoproteins [5]. Lipoprotein lipase (LPL) is an enzyme that hydrolyzes TG in TG-rich lipoproteins into one monoacylglycerol and two free fatty acids, which are subsequently taken up by muscle for utilization or by adipose tissue for storage [70]. LPL is synthesized and secreted by parenchymal cells in peripheral tissues and attached to luminal surface of capillary endothelial cells where it interacts with lipoproteins in the bloodstream [70]. LPL activation requires apolipoprotein C-II, a surface component of chylomicrons, very low-density lipoproteins (VLDL), and HDL [71]. In the presence of VLDL/HDL, LPL activity was enhanced by apoA-IV when either TG emulsion or TG-rich lipoproteins from apoC-II deficient subject was used as substrate [5]. In the absence of an apoC-II source, apoA-IV no longer promoted LPL activity. Moreover, it was the lipid-free form of apoA-IV that was functional as saturating lipoprotein-unassociated apoA-IV with lipids markedly reduced its ability to activate LPL activity.

### 3.2. ApoA-IV and Reverse Cholesterol Transport

Reverse cholesterol transport (RCT) involves a series of events resulting in the transport of excessive cholesterol from peripheral tissues back to the liver for excretion in the bile and eventually the feces [72]. It begins with the transfer of cellular cholesterol within peripheral cells to extracellular HDL-based acceptors via the action of lipid transporters, such as ATP-binding cassette transporters ABCA1 and ABCG1. Cholesterol efflux in peripheral tissue can occur in a number of ways [73]: (1) Efflux to nascent, lipid-poor HDL via ABCA1; (2) efflux to mature HDL formed as a result of esterification of free cholesterol in HDL, via ABCG1; (3) efflux to mature HDL mediated through other pathways, including possibly scavenger receptor class B type I (SR-BI) and passive diffusion. Free cholesterol in HDL is converted to cholesteryl ester by an enzyme named lecithin:cholesterol acyltransferase (LCAT). This conversion provides the driving force for more free cholesterol to move along concentration gradient from plasma membrane to HDL in the extracellular space [74]. Cholesteryl ester-containing HDL is subsequently absorbed by the liver through SR-BI. Alternatively, cholesteryl ester is transferred from HDL by cholesteryl ester transfer protein (CETP) to apoB-containing lipoproteins [75], which are later delivered to and taken up by the liver via VLDL/LDL receptors [73,76].

Several lines of evidence have indicated that apoA-IV promotes cholesterol efflux in cultured cells. Using human skin fibroblasts cultured in the medium with [^3^H]cholesterol-labeled serum, apoA-IV was shown to increase cholesterol efflux in the presence of phospholipids-containing liposomes [9]. The effect of apoA-IV was comparable to that of apoA-I and apoE at physiological relevant concentrations. In adipose cells preloaded with LDL cholesterol, both artificial and native apoA-IV-containing lipoprotein particles were able to enhance cholesterol removal as a function of time and concentration [10]. Using cultured primary macrophages with preloaded radiolabeled cholesterol, HDL-sized lipoprotein particles from apoA-IV transgenic mice conferred greater ability to reduce cellular cholesterol content than those from WT mice [12]. ApoA-IV increased cholesterol efflux in HeLa cells overexpressing human ABCA1, indicating that apoA-IV participates in ABCA1-mediated cholesterol efflux [77]. Analysis using N-terminal or C-terminal deletion mutants of apoA-IV further revealed that the C-terminal domain (aa 333–376) inhibits apoA-IV’s ability to promote cholesterol efflux [78].

ApoA-IV can activate LCAT and enhance cholesterol esterification rates [8,12]. LCAT catalyzes the transfer of the acyl group from the β-position of phosphatidylcholine (a class of phospholipids) to the hydroxyl group of cholesterol, leading to the production of lysolecithin and cholesteryl ester [79]. By using purified LCAT and phospholipid/cholesterol/[4-^14^C]cholesterol-containing liposome, human apoA-IV was proven to facilitate LCAT activity [8]. ApoA-I and apoA-IV are the two most efficient co-factors for LCAT activity [80]. However, apoA-IV differs from apoA-I in terms of acyl donor. ApoA-IV was a more potent activator of LCAT than apoA-I with L-α-phosphatidylcholine substituted with two saturated fatty acids. Whereas apoA-I acted stronger than apoA-IV with egg yolk lecithin, L-α-dioleoylphosphatidylcholine and L-α-phosphatidylcholine esterified with one saturated and one unsaturated fatty acids. By incubating the plasma with [^3^H]cholesterol followed by quantification of radiolabels incorporated into cholesteryl ester, mouse apoA-IV transgenic mice showed higher cholesterol esterification rates than WT mice, indirectly supporting apoA-IV-mediated LCAT activation in vivo [12]. Structure–function analysis revealed that partial removal (Δaa 117–160) of the amphipathic α helices region of apoA-IV markedly decreased LCAT activation by apoA-IV [78]. This suggests that residues 117–160 in the amphipathic helical sequences are involved in LCAT activating function of apoA-IV and thus critical for the properties of apoA-IV in RCT.

ApoA-IV increases human plasma CETP activity in vitro [81]. In humans, CETP is central to the alternative pathway of delivering cholesteryl ester in HDL to the liver and thereby influencing the overall rate of RCT [73,82,83]. Using lipid microemulsion, the activity of CETP was determined in the presence of apoA-IV. Cholesteryl ester transfer rate was directly proportional to the amount of apoA-IV on the surface of the lipid emulsion, indicating that apoA-IV activated CETP [81]. By studying apoA-IV mutants with truncations in various α helical regions, Weinberg et al. found that, rather than specific helical region, molecular interfacial exclusion pressure accounted for apoA-IV’s function in CETP activation [84].

### 3.3. ApoA-IV and Atherosclerosis

Over-expression of apoA-IV prevents mice from atherosclerosis. Transgenic mice over-expressing human apoA-IV in the liver of C57BL/6 strain and apoE knockout (apoE^-/-^) strain showed significantly reduced atherosclerotic lesion by a mechanism that did not involve increased HDL cholesterol concentration [11]. Similarly, expressing human apoA-IV in the intestine of apoE-deficient background (apoA-IV/E_0_ mice) protected the mice from atherosclerosis without an increase in HDL cholesterol [13]. Mouse apoA-IV over-expression engendered the same effect on the formation of atherogenic diet-induced aortic lesion [12]. Three possible mechanisms have been suggested for the anti-atherosclerotic action of apoA-IV. One is the role of apoA-IV in promoting reverse cholesterol transport, which has been reviewed previously. The other two are the anti-oxidant and anti-inflammatory activities of apoA-IV, which will be described as follows.

### 3.4. ApoA-IV and Lipoprotein Oxidation

ApoA-IV serves as an inhibitor of lipoprotein oxidation [12,13,85,86]. Oxidation of lipids in low-density lipoproteins (LDL) is a key stage in the initiation of atherosclerosis [87,88,89]. LDL enters into subendothelial space and is oxidized by mechanisms involving free radicals and/or lipoxygenases [87]. Free radicals can come from several sources: Free metal ion, low-molecular weight complexes of metal ions, superoxide, thiols, peroxynitrite, or nitric oxide [87]. Lipoxygenases catalyze the oxidation of polyunsaturated fatty acids in lipids [90]. Oxidized LDL attracts monocytes to migrate across epithelium and into artery wall, where the cells differentiate into macrophages. Macrophages take up oxidized LDL via scavenger receptors and become foam cells, a hallmark of atherosclerotic lesion. Macrophages also secrete myeloperoxidase, an enzyme catalyzing the formation of reactive oxygen intermediates that also triggers LDL oxidation. Using fasting lymph to mimic interstitial fluid in the subendothelial space, Qin et al. determined that metal ion (Cu^2+^)-induced oxidation of lymph lipoproteins or purified LDL was inhibited by rat apoA-IV in a dose-dependent manner [85]. Meanwhile, macrophage-mediated oxidation of lymph lipoproteins was reduced by apoA-IV. The antioxidant property of apoA-IV was also indicated by the finding that apoA-IV/E_0_ mice had decreased LDL aggregation and less oxidized LDL than WT [13]. Given that oxidized LDL induces the migration of monocytes to artery wall, Cohen et al. evaluated the degree of LDL oxidation by counting the number of transmigrating monocyte in a co-culture model of the artery wall [12]. Using this co-culture system, consisting of a monolayer of endothelial cells, smooth muscle cells, and an extracellular matrix in between, they found that LDL treatment stimulated a five-fold increase in monocyte migration compared to mock treatment. When LDL treatment was combined with HDL particles from apoA-IV transgenic mice, monocyte migration was reduced to a greater extent than LDL with HDL particles from control mice, implying the inhibitory function of apoA-IV on LDL oxidation. Using VLDL and apoA-IV purified from human plasma, human apoA-IV also prevented Cu^2+^-induced oxidation as well as 2,2’-Azobis(2-amidinopropane) dihydrochloride-induced oxidation of VLDL [86].

ApoA-IV regulates intracellular glutathione redox balance and thus mitigates oxidant-induced apoptotic cell death [91]. In mitotic competent, undifferentiated pheochromocytoma (PC12) cells, pretreatment with apoA-IV attenuated tert-butyl hydroperoxide (TBH), or thiol oxidant diamide-induced apoptosis. Oxidant-induced apoptosis is consistently associated with oxidant-induced glutathione (GSH)/glutathione disulfide (GSSG) imbalance [92]. Decreased GSH to GSSG ratio precedes the activation of mitochondrial apoptotic pathway, which involves loss of mitochondrial integrity, mitochondrial cytochrome c release, and cysteine protease (caspase)-3 activation [93]. GSH predominantly exists in the reduced thiol form, and is oxidized to GSSG under oxidizing conditions. Cellular GSH homeostasis is maintained through three means [93], de novo synthesis of GSH, uptake of GSH from exogenous resources across plasma membrane, and catalytic reduction of GSSG by GSSG reductase, which utilizes NADPH generated from the action of glucose-6-phosphate dehydrogenase (G6PD). The Kalogeris group discovered that ApoA-IV decreased TBH-elicited GSSG production and increased GSH levels in PC12 cells [91]. ApoA-IV did not affect de novo synthesis of GSH, as blockage of this pathway by DL-Buthionine-(S,R)-sulfoximine did not alter apoA-IV anti-apoptotic function. ApoA-IV did not change cellular GSSG reductase activity, either. Nonetheless, it stimulated a 10-fold increase in G6PD expression and conferred G6PD-dependent protection against oxidant-mediated apoptosis.

ApoA-IV acts as an anti-inflammatory agent to alleviate experimental colitis. Dextran sulfate sodium (DSS) treatment induces inflammation and acute colitis in mice. ApoA-IV substantially postponed the onset, and diminished the severity and extent of inflammatory response elicited by DSS [94]. ApoA-IV^-/-^ mice were more susceptible to DSS-induced inflammation, whereas exogenous administration of apoA-IV to apoA-IV^-/-^ in mice reversed the outcome. DSS is thought to trigger mucosal injury and confer toxicity on epithelial cells, leading to the recruitment and activation of inflammatory cells, upregulation of inflammatory mediators, and eventually the development of severe colitis [95]. ApoA-IV could significantly reduce DSS-induced leukocyte and platelet adhesive interactions, an important initial step for recruiting leukocyte to the site of injury or infection. Stimulated endothelial cells or activated platelet at the site of injury express P-selectin, a protein belonging to selectin family of cell adhesive molecules, at the plasma membrane. The interaction between P-selectin and P-selectin glycoprotein ligand-1 (PSGL-1) expressed on the cell surface of leukocytes guides the tethering and rolling of leukocytes on the endothelial cells as well as heterotypic platelet aggregation on the leukocytes [96]. The inhibitory role of apoA-IV on leukocyte and platelet adhesion was likely attributed to the downregulation of P-selectin.

### 3.5. ApoA-IV Inhibits Platelet Aggregation and Thrombosis

As the most abundant integrin in platelet, αIIbβ3 is required for platelet aggregation [97,98]. Integrins are ubiquitous transmembrane α/β heterodimeric receptors that mediate cell-to-cell and cell-to-extracellular matrix interactions upon binding to respective ligands [99]. They are present in platelets to support platelet adhesion to the extracellular matrix proteins [100,101,102]. In normal circulation, platelets exist in the non-adhesive “resting” state and become activated upon vascular injury, under which conditions platelets are exposed to immobilized adhesive proteins in the extracellular matrix or soluble platelet agonists [98]. Platelet activation leads to the conformational change of αIIbβ3 integrin followed by increased binding affinity to its ligand, such as fibrinogen (Fg), fibronectin, von Willebrand Factor (VWF), and many matrix proteins containing arginine-glycine-aspartic acid-like sequences. The ligands, such as Fg, mediate platelet adhesion and aggregation by bridging αIIbβ3 integrin in adjacent platelets [97,98]. It also triggers downstream intracellular signaling pathways, resulting in platelet spreading, granule secretion, stable adhesion, and clot retraction. Integrin αIIbβ3-mediated platelet aggregation is indispensible for hemostatic thrombi formation, which is used to stop bleeding and preserves vascular integrity and function. However, when it is dysregulated, thrombosis occurs, obstructing blood flow in the blood vessels and increasing the risk of heart attack or stroke.

ApoA-IV negatively regulates αIIbβ3-mediated platelet aggregation and thrombosis [14]. Using recombinant apoA-IV and platelet-rich plasma isolated from apoA-IV^-/-^ mice and apoA-IV transgenic mice, apoA-IV was proven to prevent Fg-dependent and Fg/VWF-independent platelet aggregation in vitro [14]. This is likely mediated through blockage of αIIbβ3-ligand interaction by apoA-IV. ApoA-IV also adversely affected αIIbβ3-mediated granule release and indirectly inhibited P-selectin expression on the platelet surface. Structure–function analysis revealed that the N-terminal region (aa 1–38) of apoA-IV was critical for apoA-IV binding to αIIbβ3 and inhibition of platelet aggregation, whereas the C-terminal region (aa 336–376) negatively modulated apoA-IV binding to αIIbβ3 through inter-molecular interactions [14]. Moreover, two conserved aspartic acids at position 5 and 13 at the N-terminus were potential recognition sites of αIIbβ3 and necessary for apoA-IV inhibitory function. Since platelet aggregation is a prerequisite for thrombosis, apoA-IV was shown to attenuate thrombus growth and promote thrombus dissolution in vitro and in vivo [14]. The inhibitory effect of apoA-IV on thrombosis is similar to that of aspirin and clopidogrel, two well-established medications to reduce blood clot formation. In addition, apoA-IV was able to prevent postprandial platelet hyperactivity induced by high-fat diet consumption in vivo [14].

### 3.6. ApoA-IV and Glucose Metabolism

ApoA-IV improves glucose homeostasis by promoting insulin secretion at high levels of glucose [15]. Under high-fat diet feeding, apoA-IV^-/-^ mice showed reduced insulin secretion and impaired glucose tolerance. By injecting recombinant apoA-IV into apoA-IV^-/-^ mice, compromised insulin secretion was rescued and glucose tolerance was improved. In addition, injecting apoA-IV into diabetic KKAy mice caused the similar glucose-lowering effect, suggesting apoA-IV had the ability to modulate glucose homeostasis. β-cells in pancreatic islets are responsible for glucose-stimulated insulin secretion [103]. After food ingestion, circulating glucose enters β-cells through passive diffusion and undergoes catabolism to generate ATP. The change in cellular ATP/ADP ratio resulting from elevated glucose metabolism triggers the closure of ATP-sensitive K^+^ (KATP) channel and depolarizes the plasma membrane, leading to the activation of Ca^2+^ channels. The influx of ionized Ca^2+^ into the cytoplasm eventually stimulates the exocytosis of readily releasable insulin-containing granules [104]. Apart from KATP channel-dependent pathways, other secondary messengers, such as cyclic adenosine monophosphate (cAMP) and diacylglycerol (DAG), can augment insulin release in the presence of ionized Ca^2+^ [104]. Recombinant apoA-IV directly enhanced insulin secretion in isolated mouse pancreatic islets under conditions of high glucose but not low glucose. This enhancement involved the regulation of cAMP levels, which affects Ca^2+^-dependent insulin secretion. However, it remains unresolved how apoA-IV enhances cAMP pathway. We speculate the existence of an apoA-IV receptor, to which apoA-IV binds and initiates downstream signaling pathway that results in elevated cAMP levels.

ApoA-IV promotes glucose uptake in mouse adipocytes via PI3K-Akt signaling pathway [18]. In the absence of insulin, glucose transporter Glut4 in peripheral cells recycles slowly between the plasma membrane and intracellular vesicular compartments, where most Glut4 resides [105]. In response to increased blood glucose levels, insulin is secreted from pancreatic islets, circulates to peripheral cells, and stimulates the enrichment of Glut4 at the plasma membrane, which facilitates the uptake of glucose. The translocation of Glut4 is mediated through a complex cascade of signaling events [106]. Insulin binds to insulin receptor on the plasma membrane of peripheral cells and activates the tyrosine kinase domain of insulin receptor, which phosphorylates the tyrosine residues of intracellar substrates, including the insulin receptor substrate family (IRS1-4). Phosphorylated IRS interacts with and activates the phosphatidylinositol 3-kinase (PI3K) signaling pathway to generate the lipid product phosphatidylinositol 3,4,5-triphosphate (PIP3), which subsequently activates phosphoinositide-dependent kinase 1 (PDK1). Activated PDK1 then phosphorylates protein kinase B (also known as Akt) and initiates Akt signaling. PI3K/Akt signaling pathways play essential roles in Glut4 translocation and glucose uptake [106]. Recombinant apoA-IV treatment on WT mice improved glucose uptake significantly in cardiac muscles, visceral white adipose, and brown adipose tissues [18]. In contrast to WT, apoA-IV^-/-^ mice had lower glucose uptake rates in tibialis anterior, extensor digitorum longus, and soleus muscles, indicating that apoA-IV contributes to efficient glucose uptake in muscles [18]. In 3T3-L1 adipocytes, apoA-IV treatment upregulated Glut4 translocation and glucose uptake in the absence of insulin [18]. ApoA-IV stimulated the expression of Akt as well as the phosphorylation of Akt in a time- and dose-dependent fashion. Blocking PI3K signaling via wortmannin abolished apoA-IV-stimulated Akt phosphorylation and decreased glucose uptake, suggesting PI3K signaling might contribute to the activation of Akt by apoA-IV. Together, apoA-IV appears to act independently from insulin to trigger PI3K/Akt signaling in adipocytes. Further studies are needed to test whether apoA-IV’s action is mediated through insulin receptor.

ApoA-IV suppresses hepatic gluconeogenesis through interaction with the nuclear receptors, nuclear receptor subfamily 1 group D member 1 (NR1D1) and nuclear receptor subfamily 4 group A member 1 (NR4A1) [16,17]. During short-term fasting, liver generates and releases glucose into the bloodstream mainly through glycogenolysis. During prolonged fasting when glycogen has already been depleted, liver produces glucose via de novo synthesis (gluconeogenesis) using lactate, pyruvate, glycerol, and amino acids as substrates [107]. Based on these substrates, a series of enzymatic events occurs during gluconeogenesis to synthesize glucose [107]. Among them, the conversion of oxaloacetate to phosphoenolpyruvate by phosphoenolpyruvate carboxykinase (PEPCK) is a key step. Another rate-limiting step is the dephosphorylation of glucose 6-phosphate by glucose-6-phosphatase (G6Pase) to release glucose. In apoA-IV^-/-^ mice, both G6Pase and PEPCK gene expressions were significantly higher than those in WT mice. In primary hepatocytes, recombinant apoA-IV treatment reduced G6Pase and PEPCK mRNA levels. The suppression of gluconeogenic gene expression by apoA-IV was mediated through NR1D1, also known as Rev-erbα, which is a nuclear receptor and transcription factor that regulates glucose homeostasis by repressing the expression of hepatic gluconeogenic genes including PEPCK and G6Pase [108,109]. ApoA-IV was shown to interact with NR1D1 in the nucleus of human HepG2 cells. With NR1D1, apoA-IV was bound to the RORα response element (RORE) of the human G6Pase promoter region to initiate transcriptional repression. Also, apoA-IV stimulated NR1D1 expression in mouse primary hepatocytes as well as HepG2 and HEK-293 cells [16]. In conclusion, apoA-IV interacted with NR1D1 to downregulate gluconeogenic genes and thus reduced the release of hepatic glucose to the bloodstream. NR4A1 was shown to interact with apoA-IV in vitro as well [17]. ApoA-IV–NR4A1 complexes colocalized near the RORE element of the human G6Pase promoter to reduce transcriptional activity. ApoA-IV increased NR4A1 expression in primary mouse hepatocytes as well as in the mouse liver [17]. Therefore, NR1D1 and NR4A1 may serve complementary roles in apoA-IV-mediated suppression of hepatic gluconeogenesis.

### 3.7. ApoA-IV and Food Intake

Inhibition of food intake by apoA-IV is mediated centrally in rats. It was reported that intracerebroventricular (ICV) injection of recombinant apoA-IV significantly and dose-dependently suppressed food intake [19]. By contrast, ICV administration of anti-rat apoA-IV serum enhanced food intake, even during the light cycle when rats usually do not eat. Later, both apoA-IV mRNA and proteins were detected in rat hypothalamus [39], where the signals for regulating food intake and energy homeostasis are integrated [110]. In particular, apoA-IV protein was present in the arcuate nucleus, ventromedial hypothalamic (VMH), paraventricular and dorsomedical areas of the hypothalamus, as well as the solitary tract of the brainstem [40]. The anorectic effect of apoA-IV mostly occurs in the central nervous system because circulating apoA-IV could not cross the blood–brain barrier [40]. Although peripheral administration of apoA-IV also suppressed food intake, the satiation signal was transmitted to the brain via intact vagus nerve [111]. Recently, the activation of hypothalamic PI3K signaling pathway has been implicated in leptin- or insulin-induced anorexia [112,113,114]. Activated PI3K triggers the phosphorylation of Akt and subsequent downstream signaling. In cultured primary hypothalamic neurons, apoA-IV increased the levels of phosphorylated Akt in a time- and dose-dependent manner [20]. This effect was abrogated by pre-incubation with PI3K inhibitor LY294002, suggesting apoA-IV stimulated PI3K-mediated Akt phosphorylation. The satiation effect of apoA-IV through PI3K/Akt signaling pathway was recapitulated in rats when ICV administration of apoA-IV was performed [20]. More specifically, the brain area responsive to apoA-IV action is VMH. In high-fat diet-induced obese rats, apoA-IV action on feeding was blunted and PI3K/Akt pathway was impaired in VMH, implying that a defective PI3K/Akt pathway in these obese rats may be responsible for reduced anorectic effect of apoA-IV [20].

## 4. ApoA-IV Interacting Partners

### 4.1. ApoA-IV Binding to Cell Surface of Several Cell Lines

ApoA-IV, together with apo A-I, A-II, C-I, C-II, CIII, and E, are major protein components of HDL [115]. HDL is closely involved in reverse cholesterol transport by mediating cholesterol efflux from peripheral cells and transporting excessive cholesterol to the liver [116]. In adult bovine aortic endothelial cells (ABAE) to which human HDL binds, free apoA-IV as well as apoA-I were able to bind to and displace HDL on the cell surface [117]. This binding was enhanced in the presence of 25-hydroxycholeterol, which induces a 3- to 10-fold increase in HDL binding sites at the surface of vascular endothelial cells [118]. In mouse adipose cells where apoA-IV-phospholipid complexes promote cholesterol efflux, apoA-IV bound to specific cell surface sites with a K_d_ value of 0.32 × l0^−^^6^ M [10]. The maximum binding capacity for apoA-IV per cell is 223,000 sites. These recognition sites appeared to be also shared by apoA-I and apoA-II. In primary rat hepatocytes, apoA-IV-phospholipid complexes could bind to cell surface and were internalized [119]. The binding site was distinct from apoE receptor sites, indicating that apoA-IV may be responsible for apoE-independent HDL uptake by the liver. A few apoA-IV binding partners have been identified, but whether apoA-IV interacts with distinct binding patterns in each target tissue to mediate its pleiotropic actions remains to be determined.

### 4.2. ApoA-IV Binding to NR1D1 and NR4A1 in Human Hepatic Carcinoma Cell Line

Rat apoA-IV was identified to interact with NR1D1 in the bacteria two-hybrid library screening using rat liver cDNA library [16]. In HepG2 cells treated with exogenous GFP-conjugated recombinant human apoA-IV, Li et al. showed that the GFP signal colocalized with endogenous human NR1D1 in the cytoplasm as well as in the nucleus [16]. Using in situ proximity ligation assay, which marked the location of interaction in the cell, apoA-IV binding to NR1D1 was confirmed in the cytoplasm and the nucleus. When nuclear proteins from HepG2 cells treated with GFP-tagged apoA-IV are used for co-immunoprecipitation, NR1D1 was pulled down by antibodies against GFP, suggesting direct interaction between NR1D1 and apoA-IV. Considering NR1D1 is a nuclear receptor, it was proposed that exogenous human apoA-IV could be taken up by hepatocytes and interacts with NR1D1 inside the cell [16]. Using proximity ligation assays and co-immunoprecipitation, NR4A1 was also shown to interact with apoA-IV in HepG2 cells [17]. The physical interaction between apoA-IV and NR1D1/NR4A1 lays a foundation for the inhibitory function of apoA-IV in hepatic gluconeogenesis, which has been reviewed in Section 3.5.

### 4.3. ApoA-IV Binding to αIIbβ3 Integrin

Human apoA-IV has recently been found to be a novel ligand for platelet αIIbβ3 integrin in the search of human plasma protein binding to activated αIIbβ3 [14]. ApoA-IV bound to the cell surface of the activated platelet but not the resting platelet, and this binding was dependent on activated αIIbβ3. Unlike other known ligands for αIIbβ3 integrin, such as Fg and fibronectin, apoA-IV interaction with the platelet did not cause its internalization. However, apoA-IV could compete with Fg for binding to αIIbβ3 integrin on the platelet. The binding affinity of apoA-IV to αIIbβ3 was about 40% of the affinity between Fg and αIIbβ3. In a reciprocal way, Fg decreased apoA-IV binding to αIIbβ3. The arginine-glycine-aspartic acid peptide, which is the canonical integrin binding motif, also inhibited apoA-IV binding to αIIbβ3 in a dose-dependent manner. However, it is unknown whether αIIbβ3-interacting apoA-IV is associated with lipids or not.

## 5. Conclusions

In summary, as a protein synthesized primarily in the intestine, apoA-IV enters into circulation through chylomicrons, affects the metabolism of lipoproteins, and exhibits anti-atherogenic or anti-diabetic effect in the circulation and peripheral tissues. ApoA-IV is able to attenuate atherosclerosis probably through three different routes (Figure 1): (1) By affecting HDL-mediated reverse cholesterol transport; (2) by reducing LDL oxidation; (3) by suppressing inflammatory responses probably through P-selectin pathway and platelet aggregation via αIIbβ3 integrin-mediated signaling. ApoA-IV influences the regulation of blood glucose levels in three different organs (Figure 2). In the pancreas, it potentiates glucose-stimulated insulin secretion via a pathway involving cAMP and downstream of Ca^2+^-mediated exocytosis of insulin-containing granules (Figure 2A). In adipose tissue, it promotes glucose uptake through activation of PI3K/Akt pathway and subsequent Glut4 translocation to the plasma membrane (Figure 2B). In the liver, it inhibits gluconeogenesis via interaction with NR1D1 and/or NR4A1 and repression of gluconeogenesis genes (Figure 2C). Central apoA-IV modulates food intake in the hypothalamus through PI3K/Akt pathway (Figure 3). Peripheral apoA-IV probably synchronizes via the vagus nerve with central apoA-IV in the regulation of food intake (Figure 3).

Although platelet apoA-IV receptor has been identified, it is tempting to speculate that apoA-IV has multiple receptors considering its involvement in a variety of biological processes and effect on many different tissues. So far, in vitro studies have implied that apoA-IV directly binds to the cell surface of endothelial cells [118], adipocytes [10], and hepatocytes [9]. As such, it seems reasonable to propose the isolation of apoA-IV receptor(s) using different cell types and tissues. Furthermore, apoA-IV exists in lipid-bound state and lipid-free form; the latter takes up more than 50% of the total circulating apoA-IV [41,42] and seems to display functional difference from the former [5]. This may add another layer of complexity to the apoA-IV–receptor interaction. Given that apoA-IV has structural and several functional similarities as apoA-I [35,120,121], it is entirely possible that apoA-IV and apoA-I bind to common receptor sites. This hypothesis is supported by the findings that both free apoA-IV and apoA-I can displace HDL on the surface of adult bovine aortic endothelial cells [117]. In addition, apoA-IV binds to apoA-I/apoA-II receptor sites on mouse adipose cells. However, these data have been collected using cell cultures in vitro. Whether apoA-IV and apoA-I share similar receptor/s in vivo awaits further investigation. After all, it is clear that identifying the apoA-IV receptor/s and related cellular pathways will become our next major advance for the understanding of the molecular function and therapeutic applications of apoA-IV.

## Figures and Tables

**Figure 1 cells-08-00319-f001:**
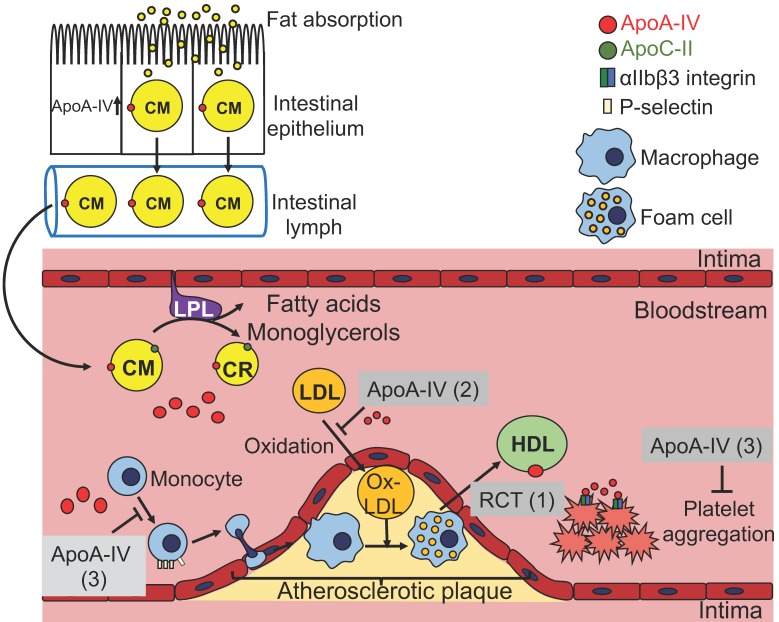
The metabolism of apoA-IV-containing lipoproteins and possible anti-atherogenic mechanisms related to apoA-IV. Upon fat absorption, apoA-IV is synthesized, secreted together with chylomicrons (CM) and transported via lymphatic system to bloodstream. In the circulation, apoA-IV promotes the activity of lipoprotein lipase (LPL) in the presence of apoC-II. Most apoA-IV dissociates from chylomicron remnants (CR) and 25% of them are transferred to high-density lipoproteins (HDL), while the rest are circulating in lipid-free state. (1) ApoA-IV is involved in reverse cholesterol transport, likely by activating LCAT activity and facilitating cholesterol transport from foam cells or other peripheral cells to HDL. (2) ApoA-IV can reduce the production of oxidized low-density lipoprotein (Ox-LDL), which is constantly taken up by macrophages to form foam cells. (3) ApoA-IV acts as anti-inflammatory agent likely by suppressing P-selectin-mediated leukocyte and platelet adhesion to endothelial cells. It also interacts with αIIbβ3 integrin and inhibits platelet aggregation.

**Figure 2 cells-08-00319-f002:**
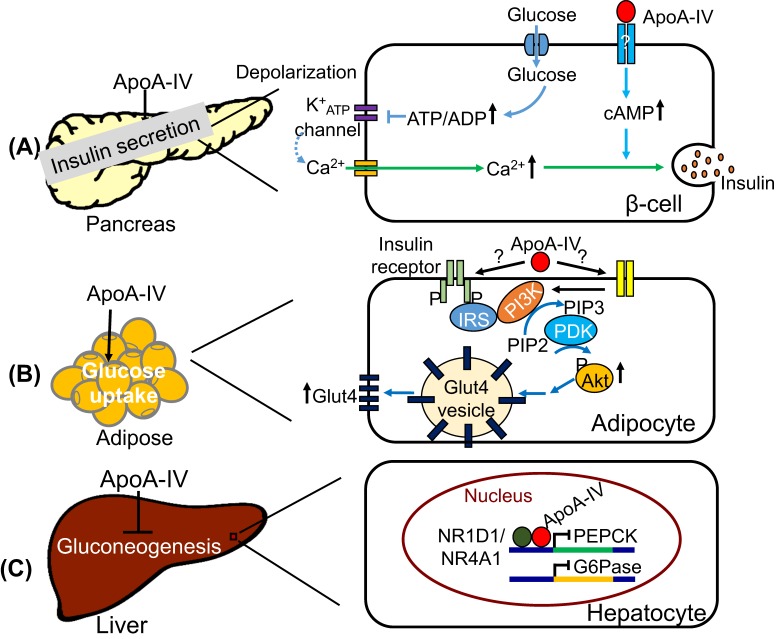
ApoA-IV improves glucose homeostasis. (**A**) ApoA-IV acts downstream of Ca^2+^-dependent insulin secretion likely via cAMP pathway. (**B**) ApoA-IV activates PI3K-mediated Akt phosphorylation and eventually promotes Glut4 translocation to the plasma membrane for glucose uptake in adipocytes. (**C**) ApoA-IV interacts with NR1D1 or NR4A1 in the nucleus to downregulate the expressions of PEPCK and G6Pase, which are involved in hepatic gluconeogenesis.

**Figure 3 cells-08-00319-f003:**
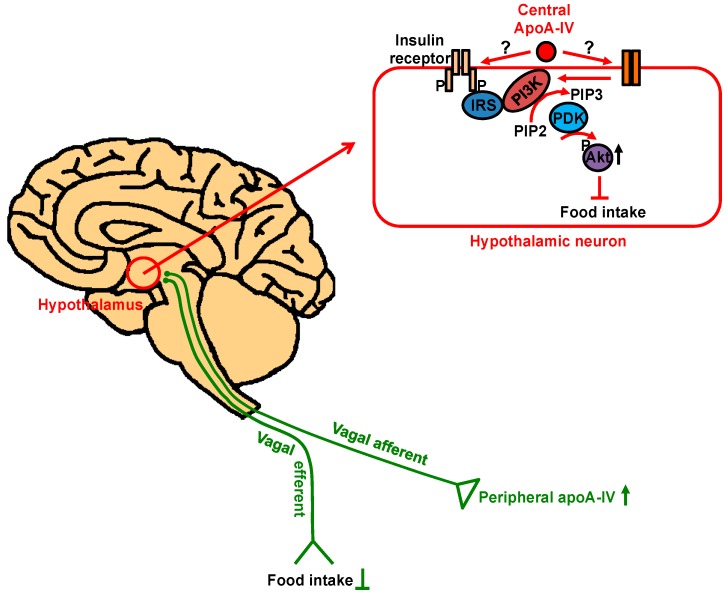
The regulation of food intake by apoA-IV. ApoA-IV expressed in the central nervous system suppresses food intake via PI3K/Akt pathway in the hypothalamic neurons. The signal elicited by peripheral apoA-IV is relayed to the brain by vagal nerve to inhibit food intake.

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
