# Peer review of "Apolipoprotein A-IV: A Multifunctional Protein Involved in Protection against Atherosclerosis and Diabetes"

_cells, 2019, doi:10.3390/cells8040319_

Round 1
Reviewer 1 Report
Excellent review on timely topic by investigator that made major contributions in this area.
Specific Comments
When discussing RCT properties of ApoA-IV should briefly comment how this biological activity likely relates to the presence of amphipathic helices.
Additional information on natural mutations/deletions of apoA-IV in patients and clinical consequences should be included.
Additional information on any association or lack of association of SNPs in apoA-IV with diabetes and or lipid levels or other clinical signs or symptoms should be included.
Author Response
We thank the reviewer for finding this topic timely. Point-by-point response is provided below:
1. When discussing RCT properties of ApoA-IV should briefly comment how this biological activity likely relates to the presence of amphipathic helices.
As suggested, this information has been added (Lines 225-237).
2. Additional information on natural mutations/deletions of apoA-IV in patients and clinical consequences should be included. Additional information on any association or lack of association of SNPs in apoA-IV with diabetes and or lipid levels or other clinical signs or symptoms should be included.
As suggested, this information has been added (Lines 65-75).
We our manuscript is now acceptable for publication.
Reviewer 2 Report
This is a really nice review that synthesizes much information on apoAIV both biochemically and in physiology. Overall it is well written, organized and references important work in the field. The figures are very nice and help to put the information reviewed together in a clear way.
Points:
Consider including the section on potenitla receptors before section 3: ApoA-IV function in cells to better set the stage for how apoA-IV may be acting.
Figure 1 - please bold the (1) (2) and (3) inside the figure so they are emphasized. Also, included the symbol (small red circle) for ApoA-IV next to or around where it is supposed to be acting would be helpful. It is used in some places, but not all.
Figure 2 - It is challenging to read the words inside the pictures of the organs, especially for adipose tissue. Incluidng a white box with black font and/or a white shadow on the letters may help.
· Section 3.2: The sentence that starts at line 170 (“A variety of pathways…”) could be moved earlier in the paragraph, for example before the sentence starting at line 164, to introduce potential mechanisms for how apoA-IV influences reverse cholesterol transport
· Paragraph starting at line 262 would benefit from a topic sentence or an introduction into apoA-IV’s role in the process before describing integrins
· Line 209: this section could benefit from more detail on ways LDL becomes oxidized, which could provide some ideas as to how apoA-IV could be acting to prevent LDL oxidation
· Section 4.2 could benefit from a brief explanation of the function of NR1D1 and NR4A1 and why interaction with apoA-IV is important or the implications of their interaction with apoA-IV (or reference the earlier paragraph in the paper about apoA-IV and glucose metabolism)
Minor points/edits:
Be consistent about your used of cholesterol and cholesterol ester or cholesteryl and cholesteryl ester.
· Line 14: it would be more clear to say “…apoA-IV is present on chylomicron remnants” instead of in chylomicron remnants, since apoA-IV associates with the phospholipid monolayer that surrounds lipoproteins
· Line 65: …yield “the” or “a” mature protein…
· Title for Section 3.1 could be made more specific by changing to “ApoA-IV and Lipoprotein Metabolism”
· Line 120: it would be more specific to move the proteins in parentheses to after “…produced by the gastrointestinal tract,…”
· Sentences at line 189-192 appear to be incomplete and are difficult to understand
· Line 233: glutathione
· Line 265: platelets exist
· Line 266: platelets are
· Line 291: reference missing
· Line 298: …are responsible for glucose-stimulated…”
· Line 320: intracellular
· Line 328: since glucose uptake was decreased, not abolished, this suggests apoA-IV may not be required per se for glucose uptake but instead contributes to glucose uptake
· Sentence beginning at line 347-350 could benefit from a rewording
· Lines 350-359: Reference for NR1D1 and NR4A1 data is missing
· Heading for “Section 3.6 ApoA-IV and food intake” should be in italics
· Line 379: omit “of hypothalamus” because the inclusion of “VMH” abbreviation includes this already
· Line 400: exogenous
Author Response
This is a really nice review that synthesizes much information on apoAIV both biochemically and in physiology. Overall it is well written, organized and references important work in the field. The figures are very nice and help to put the information reviewed together in a clear way.
We thank the reviewer for these comments.
Below, a point-by-point response is provided.
1. Consider including the section on potential receptors before section 3: ApoA-IV function in cells to better set the stage for how apoA-IV may be acting.
Two potential receptors for apoA-IV have thus far been described (NR1D1 in the liver, and αIIbβ3 in the platelet). However, we believe it may be premature to speculate on potential apoA-IV receptors without first reviewing carefully the physiological function of apoA-IV and apoA-IV interacting proteins. Therefore, we ask for the reviewer’s forbearance to include the section on potential receptors not before section 3, but at the end of this review.
2. Figure 1 - please bold the (1) (2) and (3) inside the figure so they are emphasized. Also, included the symbol (small red circle) for ApoA-IV next to or around where it is supposed to be acting would be helpful. It is used in some places, but not all.
As suggested, (1) (2) and (3) have been highlighted in white shadow to give more dramatic effect. More apoA-IV symbols have been added to the figure.
3. Figure 2 - It is challenging to read the words inside the pictures of the organs, especially for adipose tissue. Including a white box with black font and/or a white shadow on the letters may help.
The figure has been modified as suggested.
4. Section 3.2: The sentence that starts at line 170 (“A variety of pathways…”) could be moved earlier in the paragraph, for example before the sentence starting at line 164, to introduce potential mechanisms for how apoA-IV influences reverse cholesterol transport
This section has been revised.
5. Paragraph starting at line 262 would benefit from a topic sentence or an introduction into apoA-IV’s role in the process before describing integrins
This section has been revised.
6. Line 209: this section could benefit from more detail on ways LDL becomes oxidized, which could provide some ideas as to how apoA-IV could be acting to prevent LDL oxidation
Details have been added as suggested.
7. Section 4.2 could benefit from a brief explanation of the function of NR1D1 and NR4A1 and why interaction with apoA-IV is important or the implications of their interaction with apoA-IV (or reference the earlier paragraph in the paper about apoA-IV and glucose metabolism)
This section has been revised to include a brief explanation.
Minor points/edits:
8. Be consistent about your used of cholesterol and cholesterol ester or cholesteryl and cholesteryl ester.
We apologize for not being consistent. The text has been revised accordingly.
9. Line 14: it would be more clear to say “…apoA-IV is present on chylomicron remnants” instead of in chylomicron remnants, since apoA-IV associates with the phospholipid monolayer that surrounds lipoproteins
The text has been edited.
10. Line 65: …yield “the” or “a” mature protein…
The text has been revised.
11. Title for Section 3.1 could be made more specific by changing to “ApoA-IV and Lipoprotein Metabolism”
The text has been revised.
12. Line 120: it would be more specific to move the proteins in parentheses to after “…produced by the gastrointestinal tract,…”
The text has been revised.
13. Sentences at line 189-192 appear to be incomplete and are difficult to understand
The text has been revised.
14. Line 233: glutathione
Corrected.
15. Line 265: platelets exist
It has been changed.
16. Line 266: platelets are
It has been changed.
17. Line 291: reference missing
Reference has been added.
18. Line 298: …are responsible for glucose-stimulated…”
It has been changed.
19. Line 320: intracellular
It has been changed.
20. Line 328: since glucose uptake was decreased, not abolished, this suggests apoA-IV may not be required per se for glucose uptake but instead contributes to glucose uptake
Text has been revised.
21. Sentence beginning at line 347-350 could benefit from a rewording
It has been reworded.
22. Lines 350-359: Reference for NR1D1 and NR4A1 data is missing
It has been added.
23. Heading for “Section 3.6 ApoA-IV and food intake” should be in italics
Heading is now italicized.
24. Line 379: omit “of hypothalamus” because the inclusion of “VMH” abbreviation includes this already
It has been deleted.
25. Line 400: exogenous
It has been changed.
We thank the reviewer for helping us improve this review. We hope now the manuscript is acceptable for publication.